



# An improved database of flood impacts in Europe, 1870–2020: HANZE v2.1

Dominik Paprotny[1], Paweł Terefenko[2], Jakub Śledziowski[2]

[1] Potsdam Institute for Climate Impact Research (PIK), Member of the Leibniz Association,
P.O. Box 60 12 03, 14412 Potsdam, Germany
[2] Institute of Marine and Environmental Sciences, University of Szczecin, Adama Mickiewicza 16, 70-383 Szczecin, Poland
*Correspondence to*: Dominik Paprotny (dominik.paprotny@pik-potsdam.de)

**Abstract.** Assessing long-term trends in flood losses and attributing them to climatic and socio-economic changes requires
comprehensive and systematic collection of historical information. Here, we present flood impact data for Europe that is part
of the HANZE (Historical Analysis of Natural HaZards) database. The dataset covers riverine, pluvial, coastal and
compound floods that have occurred in 42 European countries between 1870 and 2020. The data was obtained by extensive
data-collection from more than 800 sources ranging from news reports through government databases to scientific papers.
The dataset includes 2521 events characterized by at least one impact statistic: area inundated, fatalities, persons affected or
economic loss. Economic losses are presented both in the original currencies and price levels as well as inflation and
exchange-rate adjusted to the 2020 value of the euro. The spatial footprint of affected areas is consistently recorded using
more than 1400 subnational units corresponding, with minor exceptions, to the European Union's Nomenclature of
Territorial Units for Statistics (NUTS), level 3. Daily start and end dates, information on causes of the event, notes on data
quality issues or associated non-flood impacts, and full bibliography of each record supplement the dataset. Apart from the
possibility to download the data (https://dx.doi.org/10.5281/zenodo.8410025), the database can be viewed, filtered and
visualized online: https://naturalhazards.eu. The dataset is designed to be complementary to HANZE-Exposure, a high-
resolution model of historical exposure changes (such as population and asset value), and be easily usable in statistical and
spatial analyses, including multi-hazard studies.

## 1 Introduction

Floods are the most significant disasters in Europe, causing more direct economic losses than any other natural hazard. In
terms of fatalities, it is second only to extreme temperature events (Centre for Research on the Epidemiology of Disasters,
2021). Further, the occurrence of floods is strongly regulated not only by climatic variation (in atmospheric forcing or  sea
level), but also evolving socioeconomic drivers such as reservoir capacity, dike height, land use, population growth, asset
value or use of private precautionary measures (Metin et al. 2018, Vousdoukas et al., 2018, Sebastian et al., 2019, Merz et
al., 2021). Numerous studies have carried out 'normalization' studies, in which reported flood losses are adjusted for



exposure growth (Pielke, 2021), both on European scale (Barredo, 2009, Paprotny et al., 2018b) and for individual countries (Barredo et al., 2012, Domeneghetti et al., 2015, Fuchs et al., 2015, Stevens et al., 2016, Andres and Badoux, 2019). Overall, the studies indicated a lack of upwards trend in exposure-adjusted direct economic loss in Europe, which has prompted research on further decomposing flood losses by including variations in hazard and vulnerability (Barendrecht et al., 2019,

Kreibich et al., 2019, Formetta and Feyen, 2019, Sauer et al., 2021, Mazzoleni et al., 2022) and projecting those findings into the future (Boulange et al., 2021, Steinhausen et al., 2022, Schoppa et al., 2023).

Notwithstanding the method used to analyse historical natural hazard impacts, having accurate, complete and georeferenced data is paramount (Kron et al., 2012). Most typical data sources in many analyses, especially on continental or global scales, are international disaster databases such as EM-DAT (Centre for Research on the Epidemiology of Disasters, 2023),

Dartmouth Flood Observatory (Brakenridge, 2023), and their extensions (e.g. Rosvold and Buhaug, 2021, Mester et al., 2023). Both rely on news media and reports by disaster relief organizations. Reinsurance companies also maintain databases, drawing heavily from insurance data, such as NatCatService (Munich Re, 2023) or Sigma Explorer (Swiss Re, 2023). The full datasets are not publicly available, hence less commonly used in research publications (Sauer et al., 2021). At European level, the Floods Directive mandated that preliminary flood risk assessments include "a description of the floods which have

occurred in the past and which had significant adverse impact" (European Union, 2007). The information on past floods, with widely varying degree of detail and completeness between countries, was collected and harmonized in a public dataset by the European Environment Agency (2015). Finally, several countries maintain national databases, some of which are publicly available on standalone websites (Guzzetti and Tonelli, 2004, Lang et al., 2016, ERMIS-F, 2020) or available through DesInventar database (United Nations Office for Disaster Risk Reduction, 2023). Where government-run datasets

are not available, research institutions sometimes fill the gap (Black and Law, 2004, Zêzere et al., 2014, Haigh et al., 2017, Andres and Badoux, 2019, Brázdil et al., 2021).

There are increasing efforts to collect and harmonize impact data from multiple countries (Papagiannaki et al., 2022, Kreibich et al., 2023). However, the largest resource on occurrence of past damaging floods in Europe remains the HANZE (Historical Analysis of Natural HaZards) database (Paprotny et al., 2018a). It contains 1564 events, covering 36 countries

and the period from 1870 to 2016, based on more than 300 data sources. Only events with at least one impact statistic (area inundation, fatalities, persons affected and economic loss) were included. Apart from flood event data, HANZE also contains a reconstruction of historical changes in exposure, to enable normalization of the reported impacts such as fatalities, persons affected and economic loss  Event data from HANZE has since been used to study trends and drivers of flood occurrence both in Europe and individual countries (Paprotny et al., 2018b, Zanardo et al., 2019, Sánchez-García et al., 2019, Thieken et

al., 2022) and validation of hydrodynamic models (Paprotny et al., 2020, Steinhausen et al., 2022).

In this paper, we present a comprehensive revision and update of the HANZE database on flood events. Thanks to incorporation of large variety of new data sources, almost 1000 new flood events were added, while almost 700 records were revised or amended with additional data, achieving a much higher degree of completeness than the original iteration. The study area was expanded to include all Balkan countries outside the European Union, and the dataset was extended to year

2020 for all countries. The database structure is largely unchanged, covering the location, date, impact, causes and bibliography of each event. As before, only events with significant socioeconomic impacts, for which at least one impact statistic is available, were included in the database. Finally, this paper greatly improves the documentation of all aspects of the database, compared to relatively brief description in Paprotny et al. (2018a). It should be noted that the associated exposure data in HANZE were also revised and vastly improved in a separate study (Paprotny and Mengel, 2023).

## 2 Methods

### 2.1 Data coverage

HANZE collects data on past damaging floods according to a set of criteria, which includes specific meaning of the term "flood", handling associated non-flood impacts, minimum thresholds for data completeness and significance of socioeconomic impacts, resolution of flood footprints, as well as spatial and temporal coverage. Such boundary conditions
were established not only to ensure homogeneity of the data, but also maximize the consistency with other pan-European datasets, primarily the exposure data and outputs of hydrodynamic models (Paprotny et al., 2017, 2019, Paprotny and Mengel, 2023). There are many edge cases connected to floods and their impacts, which are explained in the section in general, while specific cases in relation to particular variables recorded in the database are detailed in Appendix A.

### 2.1.1 Floods

The Floods Directive (European Union, 2007), which imposes minimum requirements on flood risk mapping in the EU, defines flood as "the temporary covering by water of land not normally covered by water", specifying that the definition includes "floods from rivers, mountain torrents, Mediterranean ephemeral water courses, and floods from the sea in coastal areas, and may exclude floods from sewerage systems." In similarity to this definition, HANZE includes four types of events:

- coastal floods (inundation by sea);
- riverine floods (inundation of relatively long duration by larger river systems);
- flash floods (encompassing only floods of short duration along small rivers);
- compound floods (co-occurrence in time and space of inundation from both sea and rivers).

In this context and this documentation, for brevity, "rivers" also includes lakes, which also occasionally overflow to cause
flooding. On the other hand, the database specifically excludes the following events:

- floods that were caused by insufficient drainage capacity of sewer systems in urban areas, when no inundation is caused by overflow from rivers;
- floods caused entirely by dam failures unrelated with a severe meteorological event;
- floods caused by geophysical phenomena such as tsunamis or *jökulhlaup* events (glacier outburst floods).



While floods of geophysical origin are rather clearly distinguishable, the other exclusions are not always easily determined. "Urban floods" can cause very severe economic damage, as in Copenhagen in 2011 (Rosenzweig et al., 2019) or Berlin in 2017 (Dillenardt et al., 2022), but are very difficult to model compared with riverine floods. They are particularly beyond the capacity of pan-European flood hazard models (Dottori et al., 2022), which is the primary intended use case of the dataset. As for dam failures, they happen during extreme floods, but sometimes the only cause of the disaster is a structural failure,

without direct influence of extreme hydrological conditions. Dams in this context include both reservoirs and tailing dams for mining, with the latter causing e.g. a particularly deadly disaster in Stava, Italy, in 1985 (Pirulli et al., 2017).

### 2.1.2 Inclusion criteria

As noted in the introduction, the Floods Directive obligates EU member states to collect data on past events with significant impacts, specifying them as "adverse consequences for human health, the environment, cultural heritage and economic

activity". In this context, HANZE covers floods as events which not only have caused high river flows or extreme sea levels, or even only inundation of land, but also had significant consequences. Such impacts include death or injuries to people and damage or destruction of tangible assets (buildings, other structures, contents of those, vehicles, livestock, permanent crops). Impacts only on the natural environment fall outside the scope of the study. This exclusion covers primarily extensive snowmelt floods frequent in northern Europe, which don't normally affect any assets or agriculture, as well as those coastal

storms where no inland flood was recorded beyond dunes or harbours, even though impacts to beach or erosion-protection measures can run into millions of euros (Staatliches Amt für Umwelt und Natur Rostock, 2007). Some specific exclusions in flood-related fatalities are also made, as explained in Apeendix A. Significant impacts are defined as follows in HANZE:

- At least 1000 ha (10 km$^2$) inundated;
- At least one person killed or missing presumed dead;

- At least 50 households or 200 people affected by their homes being inundated or who were evacuated;
- Losses in monetary terms corresponding to at least 1 million euro in 2020 prices and exchange rates;

To be included in HANZE, at least one of the abovementioned statistics has to be available, and must pass the defined threshold. Therefore, events that are described as causing major impacts, but without any precise quantitative information are not included. If only the area inundated is available, then the description of the flood must indicate that impacts beyond the

natural environment and into the socioeconomic sphere. A further requirement for inclusion in HANZE is the availability of information to complete certain other variables:

- Date: at least month and year has to be known, while the daily date is not mandatory;
- Type of event: the description of the circumstances should be sufficient to assign the event to the four basic types of floods (river, flash, coastal, river/coastal);

- Regions affected: the description or impact data must be sufficient to assign the impact zone using HANZE's map of subnational units (see section 2.1.3).

Other variables in the database (see section 2.3), such as detailed causes of the event, are optional, and though it was attempted to find the data for all floods, it was not always possible due to lack of sources. Priority was given to the mandatory fields which are most useful for statistical analyses of the data.

Floods often co-occur with other disasters, as the meteorological drivers are similar for other events. Therefore, reported impacts might include the effects of multiple hazards. Coastal, flash and compound floods can coincide with windstorms and thunderstorms. In hilly or mountainous areas, floods often co-occur with landslides and mudflows. Where possible, the only damages that were the results of inundation of land caused by adjacent water bodies were included, and not losses caused by strong wind, lighting strikes, hail or mass movements. Often, though, available impact information did not allow for separating impacts of coinciding hazards. In such cases, the event was included in the database, if the majority of losses was likely caused by the flooding. Co-occurrence of hazards with significant impacts were recorded in the "Notes" field (see Appendix A). If available, the amount of losses (usually the number of fatalities) was recorded as well, because at times the discrepancy in the impacts of a flood event between different data sources could be traced back to the inclusion, or not, of impacts of the co-occurring hazards.

### 2.1.3 Spatial coverage and resolution

The database covers 42 countries in Europe, as in the case of HANZE-Exposure (Paprotny et al., 2023). Included are:

- 27 European Union (EU) member states,
- four European Free Trade Agreement members (Iceland, Liechtenstein, Norway and Switzerland),
- six non-EU Balkan region countries (Albania, Bosnia and Herzegovina, Kosovo, Montenegro, North Macedonia, Serbia)
- three microstates located in Western Europe (Andorra, Monaco, San Marino),
- the United Kingdom and one of its Crown Dependencies – Isle of Man.

However, the following parts of countries are excluded:

- Canary Islands, Ceuta and Melilla (parts of Spain);
- The Azores and Madeira (parts of Portugal);
- Part of Cyprus not controlled by the Republic of Cyprus, i.e. areas controlled by the Turkish Republic of Northern Cyprus;
- All dependent or overseas territories of EU states;
- Jersey, Guernsey and Gibraltar (UK dependencies).

Additionally, for the purpose of this study:

- Data for Cyprus cover also the Sovereign Base Areas of Akrotiri and Dhekelia, and the United Nations Buffer Zone;





- The Vatican is included as part of Rome (region ITI43);
- Kosovo is shown separately from Serbia, due to its *de facto* independent status since 1999;
- The Isle of Man and Kosovo are referred to as "countries" for brevity of this paper and database.

The domain is limited to the land territory of the 42 countries, therefore impacts occurring at sea (on vessels or platforms) are not included. Disaster events that had an impact on more than one country are split per country. The limitation of the domain pertains to modern borders, therefore e.g. floods in 1930s Poland affecting the present-day territory of Ukraine are not included.

The impacts are georeferenced by recording the subnational units in which they occurred according to available sources. The subdivision of the domain into regions is largely based on the European Union's Nomenclature of Territorial Units for Statistics (NUTS). This classification has 4 levels (0, 1, 2, 3), where 0 is the national level and 3 is the finest regional division. NUTS favours administrative divisions in defining the regions, though often statistical (analytical) regions are used instead, by amalgamating smaller administrative units.

For countries in the NUTS system, the 2010 and 2021 versions of the NUTS classification, level 3, are used here (Eurostat, 2020). NUTS regions change every few years to reflect evolving administrative boundaries. All information on past events are to be recorded in two variants, i.e. according to boundaries defined in 2010 and 2021 versions. The 2010 edition was used in the original HANZE dataset and is still being used in the updated exposure model (Paprotny et al. 2018a, 2023). The 2021 edition is introduced to ensure consistency with current regional-level research in Europe. Some small countries have no NUTS subdivisions (Andorra, Cyprus, Liechtenstein, Luxembourg, Montenegro, Monaco, San Marino), while Kosovo and Bosnia and Herzegovina were still not covered by the NUTS system in the most recent 2021 edition. Therefore, subdivisions based on administrative regions of those countries were defined and coded in a manner consistent with the NUTS system. As regions for Kosovo will be formally added to the upcoming NUTS 2024 classification, our map uses the "artificial" codes from earlier HANZE iteration in the v2010 version, while the v2021 version incorporates the new official codes for NUTS 2024.

For a detailed description of the regional units in HANZE and their rendering in a high-resolution vector dataset we refer to Paprotny et al. (2023). The dataset, with simplified geometry to reduce the size of the files, was also used to render a GIS version of the HANZE database (see section 5). The HANZE map has a total of 1422 regions in version 2010 and 1443 regions in version 2021.

## 2.2 Data sources

Information for the database can be collected from all manner of published sources. They were largely available online, but this was not the requirement. Paper books were also used, while some of the online resources used in the original HANZE dataset are no longer accessible due to webpages becoming inactive or e.g. news articles being hidden behind paywalls. Individual records of events were usually compiled using multiple sources. As many sources as possible were consulted when preparing each record as a quality check. The preference was given to local sources and detailed descriptions of events,



due to numerous inconsistencies occurring especially in international disaster databases. Further, news articles report different numbers as they are often published while the event is still ongoing, therefore it was assumed that reports from a later date are more accurate. When differences between sources appeared and they couldn't be attributed to publishing date,

the source deemed more reliable should be used: a post-disaster investigation by the government or scientists is likely more trustworthy than news articles or other private publications. However, disaster databases or scientific articles often incorporate information taken from the media rather than original analysis, therefore carry over inaccurate or incomplete information. Attempts were made to trace the original source of data to remove the possibility that the citing resource misrepresented the original information.

The information on various types of impacts sometimes referred to different territorial extents, e.g. fatalities were reported for the whole country, but the persons affected were only reported by a local news outlet for a particular region. In this case, persons affected were not recorded in the database as it would be inconsistent with the fatalities. If it was possible to split the number of fatalities by regions (as defined in the HANZE map), separate entries were created according to the data availability per region. In general, an important quality check was comparing the magnitude of different types of impacts to

detect unrealistically high or low values. This is mainly visible in comparing the number of persons affected and economic value of impacts. Highly inconsistent values were not included in the database to avoid strong distortions in later statistical analyses.

Most of the data was collected from sources in national or regional languages rather than in English. International disaster databases and scientific papers provided large amounts of information in English, but they needed to be supplemented by

national sources, mainly government and news media for verification. Sources were recorded separately for each flood event to ensure transparency and enable users to quickly access more detailed descriptions of the events. A full bibliography was assembled together with a classification of the sources. Table 1 lists the different types of sources used and their frequency, together with citations of some of the most frequently consulted in making of the database. In total there were 828 sources referred to 5654 times (2.2 sources per flood event). News reports, scientific papers and government reports were most

numerous, but the international and national disaster databases were most frequently cited, constituting almost half of references. The flood and landslide database of Consiglio Nazionale delle Ricerche (2023), though only covers Italian events before 2003, is the most frequently cited resource in HANZE.





**Table 1. Data sources used in HANZE. Number of sources indicates the number of separate publications used, while number of references is the total number of times all publications were referred to in the database.**

| Type of source | Number of sources | Number of references | Notes and examples |
|---|---|---|---|
| Scientific paper | 155 | 796 | Includes book chapters and conference papers (e.g. Papagiannaki et al., 2022, Barredo, 2007, Zêzere et al., 2014, Diakakis, 2014, Olcina Cantos, 2008, Hilker et al., 2009, Mimikou and Koutsoyiannis, 1995) |
| Book | 37 | 247 | Includes MSc and PhD theses (e.g. Petrovic, 2021, Arango Selgas, 2012, Rothlisberger, 1991, Główny Komitet Przeciwpowodziowy w Warszawie, 1988, Brázdil, 2005) |
| International disaster database | 10 | 1895 | Includes compilations of national datasets (United Nations Office for Disaster Risk Reduction, 2023, European Environment Agency, 2015, 2016), news-based collections of data (Centre for Research on the Epidemiology of Disasters, 2023, Brakenridge, 2023, FloodList, 2023, Asian Disaster Reduction Centre, 2023) and intergovernmental organizations (OCHA, 2023, European Commission, 2023a) |
| National disaster database | 8 | 839 | Includes government databases (ERMIS-F, 2020, Direction Générale de la Prévention des Risques, 2023, Environment Agency, 2023) and research-based datasets (Consiglio Nazionale delle Ricerche, 2023, Pereira, 2017, University of Dundee, 2023, University of Southampton, 2023, JBA Trust, 2023) |
| Government report | 106 | 498 | Includes reports, flood risk assesments and yearbooks of government institutions and agencies, from international to local level, except the weather service (e.g. Agenzia per la protezione dell'ambiente e per i servizi tecnici, 2005, European Commission, 2023b, Krajowy Zarząd Gospodarki Wodnej, 2011, DRIEAT Île-de-France, 2022, International Sava River Basin Commission, 2014, Deutsche Gesellschaft für internationale Zusammenarbeit, 2018, Regione Siciliana, 2021, Office of Public Works, 2012) |
| Government webpage | 62 | 269 | Webpages of government institutions and agencies (e.g. Agence de l'Eau Rhône Méditerranée, 2022, DREAL Corse, 2019, DRIEAT Île-de-France, 2022, Administraţia Naţională Apele Române, 2023, Undine, 2022) |
| National weather service webpage | 37 | 209 | Webpages of government-funded meteorological or hydrometeorological agencies (e.g. Meteo France, 2023, La météo, 2023, SMHI, 2022, Met Eireann, 2018) |
| Insurer/reinsurer publication | 53 | 239 | Annual reports, websites and incidental publications of insurance and reinsurance companies (e.g. Munich Re, 1999, Finans Norge, 2023, Consorcio de Compensación de Seguros, 2022, Aon Benfield, 2017) |
| News report | 239 | 336 | Press and other media reports, including compilations of news report (e.g. Enviroportal, 2013, El Pais, 1983, Thomas, 1983, Times of Malta, 2019) |
| Wikipedia page | 79 | 183 | Includes summary articles on multiple events (e.g. Wikipedia, 2023a) and on individual events (e.g. Wikipedia, 2023b) |
| Private webpage | 42 | 143 | Includes webpages by private individuals and institutions, typically about meteorology or history (e.g. Meteo.gr, 2023, Meteo Paris, 2023, Sturmarchiv Schweiz, 2023, Belgorage, 2023) |





### 2.3 Database contents

A summary of all fields recorded in the database is provided in Table 2. Providing information for all fields is required for

each event, unless otherwise noted in the table under field "Mandatory?". This is relevant as the lack of mandatory

information excludes an event from the database (see section 2.1.2), while the lack of information in an optional field does

not. Some text fields can only use codes from a dictionary, which is contained in a separate set of tables in the dataset,

numbered S1 to S4 (see section 5). Detailed definitions of each field and discussion on the treatment of non-standard cases,

are provided in Appendix A.


**Table 2. List of fields in the HANZE database.**

| Variable | Short description | Mandatory? | Field type | Permitted values |
|---|---|---|---|---|
| ID | Unique event identifier | yes | integer | 1…9999 |
| Country code | Two-letter country code | yes | string | Codes from Table S1 |
| Year | Year of the event | yes | integer | 1870…2020 |
| Country name | Country name | yes | string | Names from Table S1 |
| Start date | Daily start date | yes | date | 1.1.1870…31.12.2020 |
| End date | Daily end date | yes | date | 1.1.1870…31.01.2021 |
| Type | Detailed type of event | yes | string | River, Flash, Coastal, River/Coastal |
| Flood source | Name(s) of rivers, lakes or seas which caused the flooding | if available | string | Free text, with names delimited by semicolon |
| Regions affected (v2010) | Regions were human or economic losses were reported, at NUTS3 level (version 2010) | yes | string | Codes from Table S2, delimited by semicolon |
| Regions affected (v2021) | As above, but at using NUTS version 2021 | yes | string | Codes from Table S3, delimited by semicolon |
| Area inundated | Area inundated (km²) | at least 1 field on impact magnitude has to be recorded | integer | 1…inf |
| Fatalities | Persons killed, incl. persons missing and presumed dead | | integer | 0...inf |
| Persons affected | Number of people evacuated or whose houses were flooded | | integer | 1…inf |
| Losses (nominal value) | Direct losses to assets in monetary terms in the currency and prices of the year and location of the flood event | | integer | 1…inf |
| Losses (original currency) | Three-letter code of currency in which nominal losses are recorded | | string | Codes from Table S4 |
| Losses (2020 euro) | Direct losses to assets in monetary terms converted to euro in 2020 prices using the gross domestic product deflator | if nominal losses are recorded | integer | 1…inf |
| Cause | The meteorological causes of the event | if available | string | Free text |
| Notes | Other relevant information or notes on issues with the data | if applicable | string | Free text |
| References | List of publications and databases from which the information was obtained | yes | string | Free text, with citations delimited by semicolon |
| Changes | Indicates changes for HANZE v1 | if applicable | string | Corrected, Updated, New |



## 3 Results

The HANZE database contains a total of 2521 flood events. Almost 49% of events are flash floods, followed by riverine

floods (46%), with coastal and compound floods constituting only 4% and 2%, respectively. In this section we analyse the distribution of flood events and fatalities. We do not analyse here the other impact variables, as due to lower completeness than fatalities, which are available for 99% of events. Analysing trends and distribution of impacts needs to consider the changing availability of data over time and space as well as long-term trends in exposure. Such analysis, as in Paprotny et al. (2018b) for HANZE v1, is out of scope of this study.

### 3.1 Temporal distribution

The contents of HANZE are presented in Fig. 1 and 2 according to their temporal distribution using the start date. All graphs show the flood events split by type, indicating the number of events (Fig. 1) and flood fatalities (Fig. 2). The number of events over time is strongly influenced by data availability, strongly improving from c. 1950 and then again from c. 1990. Nonetheless, it is relatively complete in major flood events, with no increase in flood fatalities (not adjusted from changes in

exposure or vulnerability). The distribution of events by month is more consistent and clearly follows an annual cycle from a minimum in April to a maximum in October. Coastal floods in the North Sea in 1953 and 1962 are clear outliers in the temporal distribution.

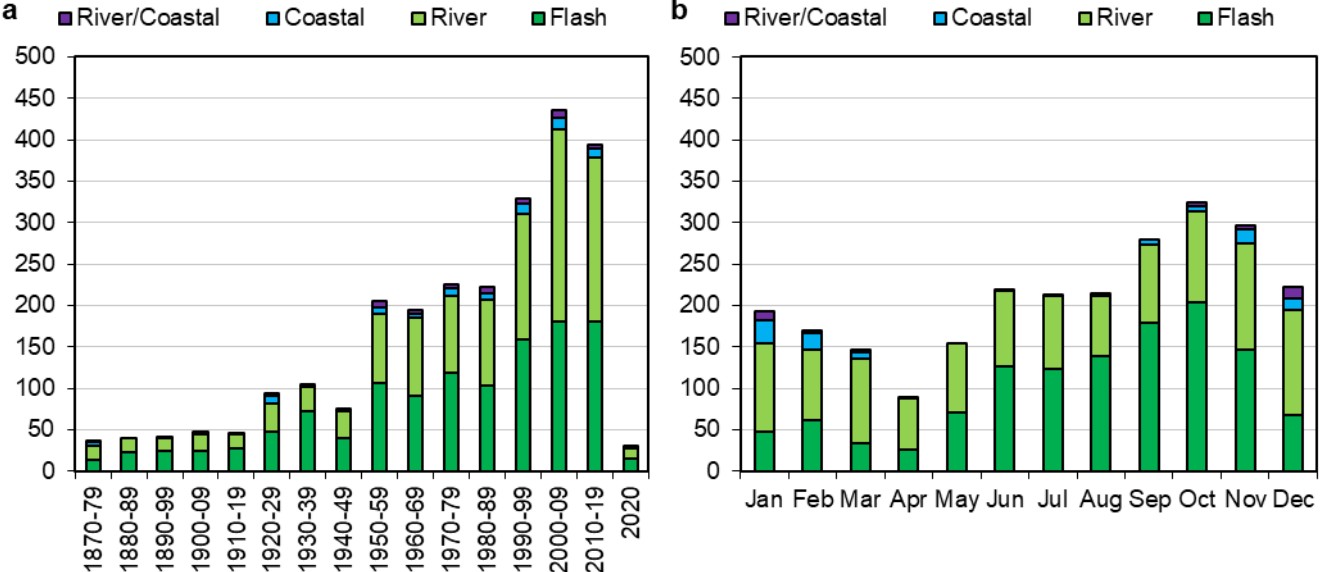

**Figure 1. Number of flood events in HANZE database by decade (a) and by month (b), 1870–2020.**

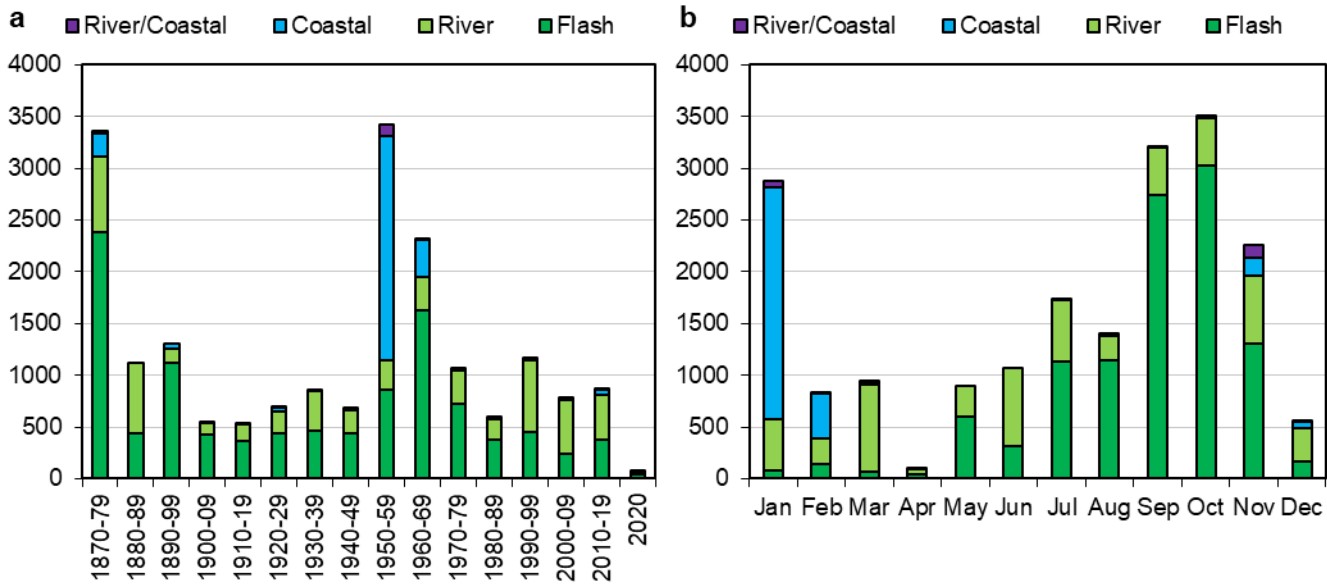

**Figure 2. Number of reported flood fatalities in HANZE database by decade (a) and by month (b), 1870–2020.**

### 3.2 Spatial distribution

HANZE contains at least one flood event for 40 out of 42 countries in the domain. With the most comprehensive sources, Italy has the highest number of events by far – 743 compared to 304 in Spain, 230 in France and 152 in the United Kingdom (Fig. 3). However, Spain has a slightly higher total number of reported fatalities (4395) than Italy (4167). Large numbers of fatalities were also recorded in France (2130), the Netherlands (1919), Germany (1325), though in the case of the Netherlands almost all fatalities are due to a single event (the 1953 North Sea flood). By contrast, very few floods were

recorded in most of northern Europe, due to the dominance of long, but not intense snowmelt floods and low population density limiting the chance of significant socioeconomic impacts. In southern Europe, flash floods strongly dominate, while in other parts of the continent slow-onset river floods are more frequent. Coastal floods form a noticeable share of events only in a few countries, located primarily along the North Sea. Finally, compound (river/coastal) floods can be found mostly along the coasts of the open Atlantic Ocean and western part of the Mediterranean Sea.

The distribution of events, of course, is affected by data availability, which is highly uneven between countries. At region level within individual countries, there is rather limited bias. The data by region (Fig. 4) show important differences in distribution of events due to occurrence of major features. In Spain, France and, to a lesser extent, in the United Kingdom the events are concentrated along the coastlines, which is not less related to occurrence of coastal floods than to higher precipitation compared to areas inland. More frequent flood events are also noticeable along major mountain ranges: Alps,

Dinaric Alps, Carpathians, and Sudetes.







**Figure 3. Number of flood events by type (pie charts) and total number of fatalities (cartogram), by country, 1870–2020.**







**Figure 4. Number of flood events by region (HANZE subnational units, v2010), total 1870–2020.**

## 4 Discussion

### 4.1 Comparison with other impact datasets

Most commonly used datasets on flood impacts are EM-DAT (Centre for Research on the Epidemiology of Disasters, 2023) and DFO (Brakenridge, 2023). DFO dataset starts in 1985, while EM-DAT nominally has coverage starting in 1900, though

in practice contains very few events before the database's inception in the 1980s. Excluding events without any impact data and considering only the HANZE domain, EM-DAT has 74 events before 1990, compared to merely 15 in DFO, but 18 times less than in HANZE (1305). Over the last 30 years, the difference is much smaller, about 3-fold between HANZE and the other two datasets (Fig. 5). HANZE already contains more events in the 1950s than either DFO or EM-DAT in the decade with most data (2000–2009). On the other hand, the other datasets have somewhat different inclusion criteria and

definitions of floods, with DFO aggregating impacts for multiple countries usually into one record. Additionally, they have the advantage of being continuously updated, compared to this one-off update of HANZE.

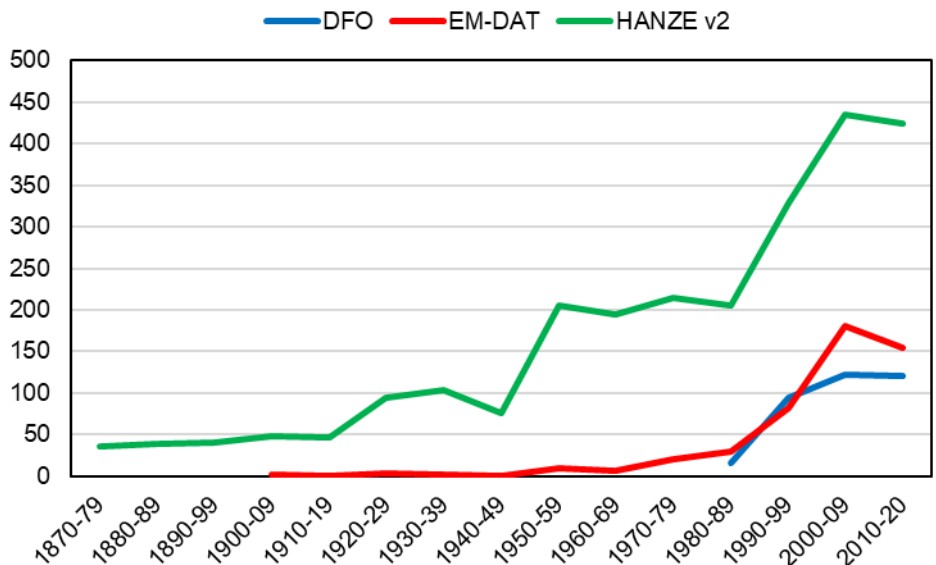

**Figure 5. Comparison in the number of floods recorded in HANZE, EM-DAT and Dartmouth Flood Observatory (DFO) datasets**
**for a comparable study area (see section 2.1.3). Excludes events in EM-DAT and DFO without any impact data.**

A more direct comparison with EM-DAT was carried out. Out of 528 EM-DAT floods in the HANZE domain until the year 2020, 479 were found to have correspondence to 468 HANZE floods. The remaining 49 floods in EM-DAT were not in HANZE for several reasons:

- 22 events without impact data in EM-DAT, and which impacts could not be added from other sources;
- 7 events below HANZE impact thresholds (section 2.1.2);
- 7 flood events outside the scope of HANZE (dam break, urban flood or hiking accident);
- 8 events verified by other sources as being primarily or entirely driven by non-flood impacts (landslide, windstorm, hail or storm at sea);
- 5 events that couldn't be confirmed by other sources or were indicated as bogus by other sources.



For EM-DAT floods found in HANZE, impact data were often different due to the use of more detailed, country-specific sources in HANZE. In cases where EM-DAT had an impact statistic, fatalities are different in HANZE in c. 50% of cases, persons affected in c. 60% of cases and economic loss in c. 40% of cases. The difference is at least partially methodological: all impacts of a multi-hazard event are included in an EM-DAT record, while the non-flood impacts were removed in

HANZE if possible. Definition of persons affected also could differ due to the use of a different multiplier in EM-DAT for flooded households, or because the population of entire administrative districts was sometimes used as "affected persons". The latter has probably the most significant influence on analysis of flood impact trends, as out of 11.6 million "total affected" persons in EM-DAT, more than half (5.9 million) is contained in just five floods. In HANZE, only about 420,000 persons are indicated as directly affected for those events. These cases are further explained in Table 3, together with a few

other examples of major differences between the datasets.

The above comparison excludes events floods classified as windstorms in EM-DAT, which happens particularly for coastal floods. Additional difference between HANZE and EM-DAT worth mentioning is the handling of economic losses. In EM-DAT, nominal losses are recorded in US dollars and then inflation-adjusted using the consumer price index for the United States. Such adjustment is incorrect as it would unrealistically assume that price levels in all countries never change relative

to the United States. In HANZE, losses are recorded in local currency and then adjusted using the gross domestic product deflator specific for each country, and then converted to euro using exchange rates in 2020.

**Table 3. Example major differences between impact data in HANZE and EM-DAT. Events in italics were excluded from HANZE.**

| Flood event | Impact type | Value in HANZE | Value in EM-DAT | Explanation |
|---|---|---|---|---|
| Italy, November 1966 | Persons affected | 88,600 | 1,300,000 | Number of persons directly affected is incomplete, with at least 88,600 evacuated (Guzzetti and Tonelli, 2004), of which 11,000 were affected in Venezia and Rovigo provinces due to a compound flood (Consiglio Nazionale delle Ricerche, 2023), which is recorded separately in HANZE |
| Genoa, Italy, October 1970 | Persons affected | 2,200 | 1,301,650 | EM-DAT value exceeds the entire population of Genoa province at the time (ISTAT, 1994); known impacts include only 2000 evacuated and 185 rendered homeless (Guzzetti and Tonelli, 2004) |
| Romania, July 1975 | Persons affected | 135,000 | 1,000,000 | A post-flood government assessment indicated 33,784 affected houses (Jurnalul National, 2005) |



| | | | | |
|---|---|---|---|---|
| Czechia, June 2013 | Persons affected | 26,438 | 1,300,000 | Value of 1.3 million was reported while the event was still in progress, likely being the total population of affected municipalities; the final number of evacuated people was only 26,438 (OCHA, 2013) |
| Bosnia and Herzegovina, May 2014 | Persons affected | 172,000 | 1,000,000 | In the early reporting of the flood, 1 million or 1.5 million affected were indicated (International Sava River Basin Commission, 2014, OCHA, 2014a), which roughly equals the population of affected regions; the actual number of flooded households was 43,000 with 90,000 persons evacuated, according to a post-flood government assessment (OCHA, 2014b). |
| Catalonia, Spain, September 1962 | Fatalities | 815 | 445 | EM-DAT value excludes missing persons, which the documentation indicates as included in "total deaths"; locally reported impacts include 441 dead and 374 missing (Martín Vide and Llasat Botija, 2000) |
| Romania, May 1970 | Fatalities | 71 | 215 | Preliminary flood risk assessment (PFRA) of Romania indicates only 71 fatalities (Administrația Națională Apele Române, 2023), while the Hungarian PFRA directly indicates no fatalities during this flood (KöTiVizIg, 2012) |
| Hungary, May 1970 | Fatalities | 0 | 300 | |
| *Romania, 1926* | Fatalities | - | 1000 | The flood (no exact date given in EM-DAT) is mentioned in Romanian literature, but no fatalities are indicated, according to a government survey (Administrația Națională Apele Române, 2009) |
| *Stava Valley, Italy, July 1985* | Fatalities | - | 329 | Failure of fluorite tailings dams of the Prealpi Mineraria mine, which was not related to any extreme hydrometeorological event; final death toll is indicated as 268 (Pirulli et al., 2017) |
| *Portugal, December 1996* | Persons affected | - | 2000 | No evidence of any extreme event in Portugal around the time of the flood indicated in EM-DAT was found by authors of the DISASTER database (Zêzere et al., 2014, Pereira, 2017) |



## 4.2 Comparison with HANZE v1

This major revision of HANZE greatly improved coverage and completeness of the dataset (Table 4). This is largely thanks to the incorporation of a bigger number of sources – 828 compared to 307 in v1. Out of 1564 events in HANZE v1, 698

(45%) were revised by revisiting the original sources or updated with new information (as indicated by "Changes" field). Apart from "Changes", one new field was introduced in HANZE v2, i.e. regions affected based on NUTS version 2021. Where there were boundary changes in regions between versions 2010 and 2021, the original data sources were revisited to correctly georeference impacts in the new map of regions.

Further 29 events of HANZE v1 were removed, for various reasons:

- 14 events removed due to merging of two HANZE records;

- 7 events removed due to stricter imposition of impact thresholds (section 2.1.2) compared to v1.

- 8 events verified, thanks to additional sources, as being outside the scope of the dataset: 3 urban floods and 5 events were impacts where driven mainly by non-flood events (landslides, whirlwinds, hail).

For transparency, the deleted records with explanation were included in the dataset in Table S7. The total number of new

events is 986, of which 192 in the six countries not included in HANZE v1. Thanks to improvement of spatial coverage, only 194 regions (v2010 map) show no flood events in HANZE v2, down from 347 previously, despite adding 70 regions through the extension of the domain. Average number of floods per region increased from 3.1 to 5.0, with 64% of regions in HANZE v1 having more events in HANZE v2. The only major exception is Portugal (Fig. 6), as originally most footprints were based on Zêzere et al. (2014), who indicated them only by upper-level administrative districts. In HANZE v2, the footprints were

revised completely using detailed geocoded impacts by Pereira (2017). Overall, verification of the impact regions from various sources caused a decline in the number of floods in 39 regions (3%), of which 25 in Portugal.

Completeness of the database has improved, though fatalities are still the only impact statistic available for almost all events. Supplementary information such as flood source and detailed cause are also more frequently available then in HANZE v1, despite lower priority given to obtaining such data.




**Table 4. Comparison between HANZE v2 (this study) and v1 (Paprotny et al., 2018a), * excluding simplified information on the cause, consistent with flood type, inserted due to lack of more precise information (see Appendix A, section A.17).**

| Category | HANZE v2 | HANZE v1 | Relative increase |
|---|---|---|---|
| Total number of events | 2521 | 1564 | 61% |
| *Number of events by type* | | | |
| Coastal | 98 | 56 | 75% |
| Flash | 1227 | 879 | 40% |
| River | 1150 | 606 | 90% |
| River/Coastal | 46 | 23 | 100% |
| *Number of events by time period* | | | |
| 1870–1949 | 484 | 414 | 17% |
| 1950–1989 | 847 | 541 | 57% |
| 1990–2016 | 1063 | 609 | 75% |
| 2017–2020 | 127 | x | x |
| *Number of regions by total impacts* | | | |
| Regions with no floods recorded | 194 | 347 | -44% |
| Regions with 1-4 floods recorded | 698 | 701 | 0% |
| Regions with 5 or more floods recorded | 530 | 304 | 74% |
| *Impact data – number of records* | | | |
| Area inundated | 394 | 157 | 151% |
| Fatalities | 2496 | 1547 | 61% |
| of which at least 1 fatality | 1589 | 1175 | 35% |
| Persons affected | 1096 | 682 | 61% |
| Economic losses | 1005 | 560 | 79% |
| *Relative completeness* | | | |
| Area inundated | 15.6% | 10.0% | 5.6 pp. |
| Fatalities | 99.0% | 98.9% | 0.1 pp. |
| Persons affected | 43.5% | 43.6% | -0.1 pp. |
| Economic losses | 39.9% | 35.8% | 4.1 pp. |
| Flood source | 65.3% | 53.1% | 12.2 pp. |
| Cause* | 41.5% | 29.3% | 12.2 pp. |






**Figure 6. Difference in total number of impacts per region in HANZE database between v2 (this study) and v1 (Paprotny et al., 2018a).**

## 4.3 Limitations and uncertainties

The dataset was compiled from many sources with varying level of reliability and completeness. The availability of information also strongly varies between countries, sometimes also within countries. The temporal bias is well noticeable,





and was analysed for the original HANZE dataset in Paprotny et al. (2018b). The analysis of the distribution of exposure-adjusted loss has shown that there was almost zero trend in the occurrence of the 20% of floods in HANZE with the biggest impacts. The lower the impact, the stronger the upward trend was noticeable. Assuming that HANZE captured all 1990–2016 flood events as well as those belonging to upper 20% by impact back to 1870, the missing losses for known events was gap-filled, and the losses for unreported events estimated by further assuming that their relative distribution, by size of impacts, should unchanged over time. Under such conditions, HANZE v1 missed an estimated 46% of flood events since 1870, though constituting only 20% of the "real" normalized impacts of floods in Europe. Even bigger impact on estimating the total normalized losses were missing impact data for known events, ranging from less than 10% for fatalities to almost 60% of the "real" impacts for the area inundated. Except for fatalities, reported losses in HANZE v1 covered less than half of the estimated true flood impacts. With the large improvement to the amount, HANZE v2 should achieve better results, though the largest additions of new events were again concentrated in the recent three decades. This indicates that more research effort is needed to obtain older sources of information, also to reduce the spatial bias between countries with comprehensive data is available (particularly France, Italy, Spain, Portugal, and Switzerland, but also Albania, Poland, Romania, Serbia and Slovenia) and those with limited or very scattered information (e.g. Baltic States, Germany, the Netherlands, Sweden, and the United Kingdom).

Further uncertainties stem from the quality of the data itself. As described in section 4.1, many differences were noticed between EM-DAT and HANZE, and that is only one example comparison. Among the largest issues are co-occurring hazards, which can strongly distort data on fatalities and economic losses. Whenever possible, attempts were made to indicate such instances and remove non-flood impacts, but often it was not possible due to the lack of detailed descriptions of the events. Other uncertainties involve data from news reports made while the events were ongoing, which might differ from the final tally made in post-flood damage surveys; problems of assigning imprecise location information to HANZE regions; or difficulty of distinguishing some floods between riverine and flash types (or coastal and compound). Most important problems are indicated in the "Notes" field, but a larger discussion on differences between sources for a particular event was avoided. The priority was given to make the database readily usable for statistical analyses and application to pan-European flood studies. Users of the database can consult the references cited individually for each event, which is a noticeable difference from many other flood datasets.

## 5 Data availability

The HANZE database is available from Zenodo repository: https://dx.doi.org/10.5281/zenodo.8410025 (Paprotny, 2023). It consists of eight comma-delimited text files and four GIS files (Table 5). Apart from the main database, a table with the full bibliography is included, as well six tables with supplementary information. Of the latter, four tables (S1–S4) include codes used in particular fields of the database (see Table 2, section 2.3), two tables (S5–S6) are used for the conversion of nominal currencies to 2020 euros (see Appendix A, section A.16), and one table (S7) contains those records from HANZE v1, which



were not included in this update (see section 4.2). The GIS files contain the information for the database, but also include a polygon per each event consisting of affected regions, in two versions (2010 and 2021 map). The geometry of the regional boundaries is simplified to reduce the file size, and the maps of all regions are also included in the supplementary data.

Additionally, the HANZE database can be viewed online at https://naturalhazards.eu/. It includes the following functionalities:

• Database: an interactive table with the most relevant fields from HANZE. The table can be sorted, searched by keywords, as well as filtered (by years, countries and types). Clicking on "Detail" for a particular event leads to a subpage in which the full HANZE record is displayed together with a map showing the location of affected regions. For better user experience, regions affected and currencies are converted from codes to full names in these event pages. Clicking on a region's name highlights it in a different colour on the map.

• Map: an interactive map which displays all affected regions for events fulfilling the search criteria (years, countries and types). On the map, each region can be clicked, which shows a pop-up indicating the number of events affecting that region within the search criteria, and a link that enables displaying the table of all events in the region. By default, the map displays a cartogram of all events between 1870 and 2020.

• List of references: a table that is sortable and searchable by keywords, containing full bibliographical details and
links to sources used in HANZE.

The webpage also provides general information on the other aspects of the HANZE database, such as the associated exposure data (Paprotny et al., 2023).

**Table 5. List of files for download from HANZE database repository (https://dx.doi.org/10.5281/zenodo.8410025). All ".csv" files**
**are comma-delimited, coded in UTF8. All ".zip" files contain a set of files constituting together an ESRI shapefile.**

| Dataset | Description | Data structure |
|---|---|---|
| *HANZE flood events database* | | |
| HANZE_events.csv | Flood event data | See Table 2 in section 2.3 for dataset structure |
| HANZE_references.csv | List of all references | No. – ID of the source<br>Name – bibliographical details<br>URL – hyperlink or DOI, if available<br>Type – type of source (see Table 1, section 2.2) |
| HANZE_events_regions_2010.zip | Flood event data as GIS file (regions v2010) | As in Table 2 in section 2.3, except:<br>- field names are truncated due to the limitation on column lengths in shapefiles; |
| HANZE_events_regions_2021.zip | Flood event data as GIS file (regions v2021) | - "Start date" and "End date" fields are spilt into separate columns for year, month, day. |



| *Supplementary data* | | |
|---|---|---|
| S1_countries_codes_and_names.csv | Country codes/names | Code – country code<br>Name – country name |
| S2_regions_codes_and_names_v2010.csv<br><br>S3_regions_codes_and_names_v2021.csv | Region codes/names, v2010 and v2021 | Code – region code<br>Name – region name |
| S4_list_of_all_currencies_by_country.csv | Data on all currencies used in the study area since 1870 | Code - country code<br>Currency - currency name<br>Code - three-letter currency code (ISO 4217, if possible)<br>Start date - date when currency first entered circulation<br>End date - data when currency was withdrawn from circulation<br>Conversion – conversion factor between new and old currency<br>Note – important annotations about the currency |
| S5_currency_conversion_rates.csv | Conversion rates applied to compute losses in 2020 euros | Code - country code<br>Currency - three-letter currency code<br>Conversion rate – rate between currency and euro |
| S6_GDP_deflators_by_country.csv | Gross domestic product deflator | Code - country code<br>1870…2020 - deflator, 2020 = 100 |
| S7_floods_removed_from_HANZE.csv | Flood events in HANZE v1, which were excluded from v2 | As in Table 2 (section 2.3), except:<br>- "Explanation" field is added instead "Changes" to detail the reason for not including the event in HANZE v2<br>- "Regions affected (v2021)" is left blank, as it was not included in HANZE v1<br>- Inflation-adjusted losses are in 2011 euros, as in HANZE v1 |
| Regions_v2010_simplified.zip<br><br>Regions_v2021_simplified.zip | Map of subnational regions used in the database, v2010 and v2021 (see section 2.1.3) | Code – region code<br>Name – region name |

## 6 Conclusions

Collection of historical flood impact data is a challenge involving searching, verifying and homogenizing information from a variety of sources. Strong differences in availability of information between countries and the bias towards recent events 415 complicate spatial and temporal analyses of the data. Nonetheless, HANZE is a significant contribution in the flood research field. With more than 2500 events spread over some 1400 regions, and containing nearly 5000 impact data points, it is by far the largest flood impact dataset available for Europe. Together with the associated historical exposure data (Paprotny et al., 2023), it is an essential step towards impact attribution of historical events beyond singular case studies. It also enables studying compound, cascading and consecutive events. The dataset also is a template for future extensions into other types of 420 hazards, supporting broader pan-European natural hazards modelling efforts.

*Code availability:* code used for aggregating HANZE data by region and generating the GIS version of the dataset is available on Zenodo: https://dx.doi.org/10.5281/zenodo.8223833

*Author contributions.* DP developed the concept, implemented the methods, collected and processed the data and acquired 425 funding. PT and JS created the online database and its visualization. All authors wrote the paper.

*Competing interests.* The authors declare that they have no conflict of interest.

*Acknowledgements.* The authors would like to thank Belinda Rhein for her assistance in data collection, as well as Matthias Mengel, Mahé Perrette and Luc Feyen for technical discussions.

*Financial support.* This research has been supported by the German Research Foundation (DFG) through project 430 "Decomposition of flood losses by environmental and economic drivers" (FloodDrivers), grant no. 449175973.

## Appendix A. Detailed definitions of fields in HANZE.

Fields contained in HANZE, briefly defined in section 2.3, are explained here in detailed.

### A.1 ID

This field provides a unique numeric identifier of each event. Floods that also occur in the original HANZE dataset from 435 2017 have IDs from 1 to 1564, while new additions have IDs from 2000 upwards.

### A.2 Country code

This field contains the two-letter country code in which the event occurred. The codes are consistent with NUTS0 codes rather than ISO 3166-1 codes. The difference between the two is that EL is used for Greece instead of GR, UK is used for



the United Kingdom instead of GB, and XK is used for Kosovo (missing in the ISO standard). The list of all codes is included in the database as Table S1.

In case of events occurring in countries which do not exist anymore (e.g. Czechoslovakia, East Germany, the Soviet Union or Yugoslavia), the appropriate successor state is assigned. If an event affected the territory of more than one successor state before the dissolution of the predecessor (e.g. Slovenia and Croatia while part of Yugoslavia), the event was recorded only if it was possible to create entries in which the impacts are recorded using the modern country divisions. In case of boundary

changes of countries existing presently, the country is assigned according to the political divisions at the time of the event. For instance, a flood in Alsace between 1871 and 1918 will be recorded as an event in Germany, even if it is presently part of France and the appropriate NUTS regions for France are assigned. Similarly, floods along most of the Oder river will be recorded under Poland after 1945 and Germany before that year.

**A.3Year**

Year in which the event occurred, based on the starting date.

**A.4 Country name**

This field contains the country name in which the event occurred, according to rules explained in section A.2.

**A.5 Start date**

Exact date on which the event started, which will preferably be the first day on which impacts were reported in the country

affected. Therefore, it can differ between countries affected by the same extreme hydrological event. If the exact daily date is not available or unclear, the best possible approximation was made and information about this was recorded in "Notes". If no information about the daily date is available, the 1$^{st}$ day of the month was inserted and information about this was recorded in "Notes". Events for which even the month is not reported were not included in the database.

**A.6 End date**

Exact date on which the event ended. If no information about the daily date is available and couldn't be approximated, the last day of the month was inserted and information about this was recorded in "Notes". The end date could possibly stretch into 2021 as long as the start date was within 2020 as well as the peak intensity of the event had occurred within 2020. However, no such case was actually recorded in the dataset.

**A.7 Type**

Type of flood event, based on the driver of impacts. "River" and "Flash" floods were distinguished according to the duration of the triggering meteorological event: if it lasted no more than 24 hours, the event was considered a "Flash" flood;



otherwise it was considered a "River" flood. Compound floods were recorded as "River/Coastal" type to avoid confusion as to the meaning of "compound" flood. This type was assigned only if the description of the event indicated that both riverine and coastal flooding contributed to the impacts. If high river flows and a storm surge merely coincided and only one of the two has caused the vast majority of losses, the event was assigned to "Coastal", "River" or "Flash" type.

## A.8 Flood source

This field records the names of the main rivers, lakes, seas or other coastal basins which have overflown to inundate land and cause impacts. Along the coastline, typically only one sea or other important nautical division (e.g. Bay of Biscay, the English Channel, the Danish Straits) will be involved. Inland, many sub-catchments are typically affected by a flood, therefore only a few of the most important rivers are listed. Multiple names are separated by semicolons.

## A.9 Regions affected (v2010)

The location of the impacts was recorded using the HANZE map of subnational units (regions) based mostly on the 2010 version of NUTS level 3 classification (see section 2.1.3 and Table S2 in the dataset). Impact at the regional level is defined the same as at country level and denotes occurrence of damages (fatalities, injuries, evacuations, damage/destruction of assets). In practice, any region reported as affected by news media will usually be recorded here. If more detailed per-region losses were reported, only regions where the vast majority (more than 90%) of losses occurred were included. Also, only if more detailed information is available, the list of regions affected included any region that recorded significant impacts defined as one of the following:

- At least 2.5 km$^2$ of land inundated;
- At least one person killed or missing;
- At least 13 households or 50 people evacuated or whose houses were flooded;
- Losses in monetary terms corresponding to at least 250,000 euro in 2020 prices.

Except for fatalities, the thresholds equal one-fourth of the event-level thresholds (section 2.1.2). In case of an event affecting several regions of a country, when the availability of impact data per region was uneven, it was split into separate HANZE events, each for a particular set of regions.

## A.10 Regions affected (v2021)

This field is recorded in the same way as the previous one (section A.9), with the difference that another map of regions is used, this time based on the 2021 version of NUTS level 3 classification, rather than the 2010 version (see section 2.1.3 and Table S3 in the dataset).



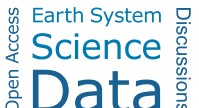

### A.11 Area inundated

The total area of land inundated during the flood event. Preferably, this included all types of land. However, more often than not only the agricultural land flooded was reported. As agricultural land is nonetheless the most typical land use in Europe, the value was recorded "as is", with no mention of this slight inconsistency in "Notes".

### A.12 Fatalities

This field records the number of people killed in connection to the event. It might involve not only the resident population, but also tourists, transients or emergency responders. Persons missing but presumed dead were also included in the total. However, certain flood-related cases of fatalities were excluded here:

- Fatalities by wave action at the coast, where persons were swept away by waves while standing on piers, waterfronts, beaches, etc. (e.g. cause of several fatalities in Ireland, see O'Brien et al., 2018);
- Fatalities by structural collapse of houses caused by heavy rainfall (e.g. several instances in Malta, see Times of Malta, 2019);
- Fatalities related to sports activities in remote areas, such as canoeing or hiking, in connection to by sudden localized flash floods, as they cannot be reliably modelled both in terms of hazard and exposure, e.g. Switzerland 1999 (21 fatalities, Sturmarchiv Schweiz, 2023), Greece 2007 (8 fatalities, Meteo.gr, 2023) or Italy 2018 (10 fatalities, FloodList, 2023);
- Fatalities that could be rather attributed to accidents (particularly vehicular) in connection to very localized bad weather, rather than directly caused by flooding (e.g. many cases reported in the UK, see Thomas, 2000).

In contrast to other fields recording impacts, the value of 0 is allowed to be recorded here if it is known for certain that the event involved no fatalities. However, in such a case at least one other field related to impacts had to be recorded. If the number of fatalities was unknown or highly uncertain, the field left blank, with "Notes" recording what is known in connection to occurrence of fatalities (such as the minimum number of known/probable fatalities).

### A.13 Persons affected

The number of persons affected is reported variously depending on the source. Here, we used either the number of persons evacuated, or the number of homes inundated. If the number of houses (or buildings, households, dwellings, or similar expressions) is reported rather than the population living in them, the number of persons was estimated by multiplying the number of houses by 4. The persons evacuated or affected by inundation typically overlap, therefore if both were reported, the higher number was used and not a summation of the figures. Similarly, the number of injured people, if available, was not added to the number of persons affected or recorded if it is the only statistic available. For major flood events, if both the number of flooded households and evacuated was available, the number of evacuations was additionally recorded in "Notes".



**A.14 Losses (nominal value)**

The monetary estimate of losses from a flood is recorded here. This includes only the direct losses to tangible assets (buildings, other structures, contents of those, vehicles, livestock, permanent crops). Losses to inventories (such as non-permanent crops or finished manufactured goods) as well as indirect losses (effects of interruptions to the normal functioning of the economy) were excluded, though in practice only in very rare instances such data were published (e.g. Chatterton et al., 2010). In this field, the numeric value of the loss estimate was inserted, while the currency to which the number refers to is recorded separately (see section A.15). The value represents the currency that was used at the location and time of the event. If the source didn't provide the loss estimate in the required currency, the value was recalculated using exchange rates at the time of the event (obtained from various economic data sources, see Paprotny et al., 2023). On rare occasions, the sources reported losses in a price level of a different year than when the event occurred, hence it was adjusted using gross domestic product deflators (see section A.16).

Insured losses are sometimes available instead of total losses, but they were not used due to limited coverage of flood insurance in most countries. Exceptions were made for compensated losses paid through comprehensive government schemes in Denmark (coastal floods only), Norway and Spain, which often exceeded alternative estimates of total losses. Such instances were indicated in "Notes".

**A.15 Losses (original currency)**

This field records the currency in which the nominal value of estimated losses was inserted in the previous field (section A.14). The currency is reported with a three-letter code, a complete list of which is provided in the dataset in Table S4. The list was adapted from Paprotny et al., 2023. For most currencies, the codes are the same as in the international standard ISO 4217, though several historical currencies were assigned artificial codes that follow a similar pattern, i.e. the first two letters identify the country and the third letter the currency.

**A16 Losses (2020 euro)**

Here, the monetary estimate of losses from previous fields (sections A.14-15) is standardized to a single currency and price level. This is done by recalculating the nominal losses to euros at 2020 price level and exchange rates. Firstly, the nominal value is converted from the local currency of the time of the event to euros using conversion rates in Table S5. Then, it is converted to the 2020 price level by applying gross domestic product (GDP) deflators from Paprotny et al. (2023). For example, a loss of 100,000,000,000 PLZ (100 billion "old" Złoty) in 1992 converts to 10,000,000 PLN (10 million Złoty), and then to 2,250,731 EUR according to Table S5. The deflator for Poland for 1992 is 16.504 (Table S6), where year 2020 equals 100, therefore:

$$\frac{2,250,731 \times 100}{16.504} = 13,637,229 \tag{A1}$$

In this example, the value of 13,637,229 would be recorded in this field.





**A.17 Cause**

Meteorological extremes observed during the event are recorded here. If known, the maximum intensity or total amount of precipitation was recorded, in millimetres per time unit, unless the event was caused by snowmelt or ice jam. In case of
coastal floods, the maximum surge heights or wind speed was provided, if available. If no specific data is available, a generic text was inserted according to the type of event: "Extreme rainfall" (intense but short duration) for flash floods, "Heavy rainfall" (less intense but of longer duration) for river floods, and "Storm surge" for coastal floods.

**A.18 Notes**

This optional field records other information that is relevant for the event, but cannot be included in any other field. Such
cases include, but are not limited to, information that:

- the daily date of the start/end of the event is not exact, but approximate or assigned to the 1[st] day of the month (see sections A.5-6);
- the impact data for all/selected fields include, or likely include, impacts of co-occurring hazards (see also section 2.1.2)
- the impact data is limited to only part of the affected area (i.e. not all affected regions);
- economic loss pertains to damages compensated through a comprehensive government scheme;
- that there are quality issues (incompleteness, low reliability) with the recorded impact data.

The field was also filled particularly if the following information was available:

- co-occurrence of related events (with significant impacts) such as landslides, mudslides, lightning strikes or hail;
- impacts of those co-occurring hazards;
- occurrence of major flood defence/control breaches (dam breaks or a large number of dike failures);
- occurrence of multiple flood waves within one event;
- estimated return period of events or other statistical information indicating the extremity of the event not recorded under "Causes" (see A.17).

Other information of high relevance and in the scope of the study, not mentioned above, were also at times recorded, but having in mind the need to keep this field concise and prioritizing the mandatory fields.

**A.19 References**

All sources that were used to compile the record of an event are listed here. They are recorded in the standard author-year format, e.g. "European Commission (2023)" or "Papagiannaki et al. (2022)". Individual sources are separated by semicolons.
The full bibliographic details of the sources are kept in a separate table in the database (see section 5), including the DOI (if available) or URL as well as classification of the type of source (as in Table 1, section 2.2).



## A.20 Changes

This field indicates whether the record differs from HANZE v1. The record is not considered changed due to addition of "Regions affected (v2021)", the correction of typos in the optional fields, or updating the list of references without modifying other data. Changes made by revisiting previously indicated sources are indicated by "Corrected", while those made by consulting additional sources are indicated by "Updated". New events are indicated by "New" and those left unchanged by a blank field.

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
