# Peer review of "An improved database of flood impacts in Europe, 1870–2020: HANZE v2.1"

_Earth System Science Data, 2023_

## Author Response (AR1)

We thank the reviewer for taking the time to analyze our paper. Below we list the reviewer's comments with our responses afterwards.

**Comment 1:** Item 2.1: It is not clear what these criteria are. What is specifically meant by the "specific meaning of the term flood"? I understand you specify the criteria later; however, it reads very confusingly to have it here without going into detail. I would suggest modifying this paragraph.

**Response 1:** the paragraph is an introduction to the section, where individual criteria are explained in the following subsections. We have modified the paragraph so that it now mentions the subsections where the criteria are discussed.

**Comment 2:** Line 103: I see it as a limitation that you only consider floods with high consequences. EM-DAT and existing datasets also focus on these high-damage floods. However, smaller-scale events can have high damages when considering their cumulative consequences. I would suggest writing about this in the limitations section.

**Response 2:** we would disagree that we only consider floods with high consequences. We can note that EM-DAT has a minimum threshold of 10 fatalities, a "call for international assistance or an emergency declaration", or 100 affected. While the first two indicate only very major floods, the third is theoretically lower than ours (200 affected), but in practice rarely applied. Also, the other database cited by the reviewer, FFEM, only considers events with fatalities. By contrast, 907 events (36%) in HANZE caused no fatalities, which amounts to almost twice the number of all EM-DAT events. We also include events with as few as 10 affected persons due to relatively low economic threshold in particular (1 million euro in 2020 prices), which enables capturing floods that impacted commercial assets or infrastructure, but spared residential zones. We added the following to the discussion:

„In a limited number of cases, this could be explained by different inclusion thresholds, but overall HANZE has less strict criteria regarding the minimum magnitude of floods that warrants adding them to the database, compared to other databases. For instance, 36% of events in HANZE had no fatalities, whereas FFEM database is limited to events leading to fatalities."

**Comment 3:** Line 112: I would mention already here that only one criterion needs to be fulfilled.

**Response 3:** We have moved the sentence before the list of criteria.

**Comment 4:** Item 2.2: This item needs better explanation. How did you find the correct sources? How can this be reproduced by other researchers? Which keywords were used to search for articles or papers in different languages? What are these local sources? Is it a local newspaper article? How were ambiguities solved? Was this done manually or with natural language processing? How many people verified this information? What is the intercoder reliability? There is a vast literature on document analysis (the method you used here, I assume) that should be referenced at least once.

**Response 4:** we have revised and extended the text to cover the questions posed by the reviewer where they were not addressed so far. We added the following:

"Each country was researched separately, first drawing information from international and national databases (if available), national or regional flood risk assessments, and then supplemented by search of online resources. Research papers were searched in repositories using the country name and the term "flood" as keywords. Using online search engines, news reports, government and private websites were uncovered using various flood-related terms, the country name or particular year/location of floods (if known from previously collected information), in both English and the national languages of the countries researched."

Further, we modified the sentence where "local" sources are mentioned for clarity: "The preference was given to country-specific sources (national databases, news articles, government reports) and those with detailed descriptions of events, due to numerous inconsistencies occurring especially in international disaster databases."

We also added the following: "To increase consistency, all entries in the database were completed and inserted by one person, even if sources were located and pre-processed by other researchers involved, in particular due to their better knowledge of a specific national language."

We also already highlight that "Sources were recorded separately for each flood event to ensure transparency and enable users to quickly access more detailed descriptions of the events." We consider the open publication of data and full detailed referencing of each event contained a particular advantage of HANZE over other studies and indication of the authors' dedication to transparency. We note that other studies do not publish detailed references (like FFEM, not to mention other international databases) or do not publish underlying event data at all (many research papers, e.g. Brázdil et al., Potential of Documentary Evidence to Study Fatalities of Hydrological and Meteorological Events in the Czech Republic. *Water* 2019, *11*, 2014. https://doi.org/10.3390/w11102014).

Finally, we are not sure what is the "literature on document analysis", as we haven't seen such literature cited in other papers on historical flood data collection. We would be happy to include any suggestions by the reviewer.

**Comment 5:** Figure 1: Why is there so little information for 2020? Did you consider the entire year?

**Response 5:** 2020 includes 30 events, which is below the average for previous decades (about 40 per year), but more than e.g. 2011, 2012 or 2017. We have modified the graph by adding 2020 to the previous decade, to avoid confusion.

**Comment 6:** Figure 3: This figure is not appropriate for color-blind people. Also, there is a lot of information overload. I suggest having two separate figures. The pie charts should be removed as they sometimes cover the entire country. It would be interesting to have information about how reliable/complete the information for each country is.

**Response 6:** we have modified the color scheme and have split the figure into two. We added a table (Table 6) to show the completeness of impact data, and a qualitative indication of the perceived completeness of flood records, per country.

**Comment 7:** Discussion: A comparison with the database on fatalities by Pagagiannaki (2022) for the countries covered in both databases should be made.

**Response 7:** we have added such a comparison to section 4.1:

„In a recent database FFEM (Papagiannaki et al., 2022), data on 1524 fatalities since 1980 in nine countries also covered by HANZE were collected. In case of two countries (France, Spain) only some parts were included in FFEM. Overall, there are more fatalities indicated in HANZE (1796) than in FFEM, though in some countries (Cyprus, Czechia, Portugal). The difference is partially methodological, as certain fatalities in FFEM were found to be outside the definition of used in this study (section A.12). Otherwise, it was the result FFEM being the only source of information that was traced – FFEM itself doesn't include enough information to fulfil all inclusion criteria (section 2.1.2)."

We added also Table 4 to support the comparison.

**Comment 8:** Line 282: What does 1305 mean here? Is it the number of events in Hanze before 1980?

**Response 8:** We have tweaked sentence to make it clear that it is the number of events in HANZE before 1990:

"Excluding events without any impact data and considering only the HANZE domain, EM-DAT has 74 events before 1990, compared to merely 15 in DFO, but 18 times less than in HANZE (1305 events before 1990)."

**Comment 9:** Figure 4: It would be nice if you could compare these results with water gauge levels over the years, not only with other databases. Can you do this for at least some of the countries? This would improve reliability.

**Response 9:** it is unclear to us how such comparison should look like. Water levels translate only partially into impacts, due to large variation in flood protection levels and exposure. Additionally, the availability of gauge data is highly heterogenous in Europe. Therefore, such a comparison would not be informative in our opinion. We are currently merging a hydrological reanalysis with gauge data and documentary records of flood impacts and non-impacts, but due to very extensive modelling work involved it is outside the scope of this publication.

**Comment 10: Figure 5:** Please add Hanze V1 here too.

**Response 10:** We will include HANZE v1 in the revision.

**Comment 11:** The direct comparison with EM-DAT is not done in a fair way since the criteria for inclusion in EM-DAT are different. If you consider the same criteria as EM-DAT, how many more events does your database have? Please add two sentences with this comparison.

**Response 11:** The thresholds in EM-DAT are both higher (fatalities) and lower (persons affected) than ours, and additionally include the troublesome criterion of "call for international assistance or an emergency declaration". Consequently, it is not possible to fully

homogenize the criteria. When filtering events from both datasets using only the first two criteria, HANZE still includes three times more events than EM-DAT. As we mention already in section 4.1, we only found seven floods (out of 528) in EM-DAT that are not included in HANZE due to falling outside our criteria. We modified the discussion in section 4.3 to mention the threshold aspect (see Response 2).

**Comment 12:** 4.2: The revision of the entries in Hanze 2 highlights how variable the methodology is and how much it depends on personal interpretation and different sources. This limitation should be highlighted.

**Response 12:** we would disagree with such an interpretation of the revisions made to the database. The main cause of revisions is the incorporation of a vastly greater number of sources (almost three times more). That an individual record was "revised" often doesn't mean that previous information was changed, but added where it was previously missing. A revision was indicated when changes were made to any field, even to non-mandatory supplementary information (such as precipitation amount or notes). Availability of sources changes constantly, whether it's a new research paper on flood events in Malta or a newly digitized 1960s post-flood government report from Austria. Each adds new level of detail and precision that changes the previous assessment that was sometimes based on very limited or outrightly incorrect secondary information, frequently those included in international databases.

**Comment 13:** Line 355: It would be important to have an assessment of the completeness and reliability for each country, even if it's qualitative. This information is crucial for people using this dataset and taking it for granted.

**Response 13:** assessing the "true" number of floods per country is not possible. As noted in response 9, we are trying to partially address this by employing extensive model reconstruction of past floods, which falls outside the scope of this paper. We added a table (Table 6) to show the completeness of impact data, and a qualitative indication of the perceived completeness of flood records, per country.

**Comment 14:** Line 382: The references provided are not that useful, in my view. For example, how do I know which paper is (UNDRR 2020)? It would be great if you provided full references in the Hanze_events.csv dataset (not separate as it is now).

**Response 14:** HANZE database follows the standard academic citation format. As the list of references is alphabetical, it is not difficult to find UNDRR (2022) there, as in the bibliography of this research paper.

**Anonymous Referee #2**

We thank the reviewer for taking the time to analyze our paper. Below we list the reviewer's comments with our responses afterwards.

**Comment 1:** How is the flood extent validated? Is it done considering the DFO database? I highly recommend to validate the flood extent from your database with the ones reported in the Global Flood Database by Tellman et al. (2021)

**Response 1:** The flood extent is based directly on the documentary sources that are cited per each flood. We disagree with the reviewer that HANZE should be validated with the Global Flood Database. This is because satellite-derived footprints are not direct observations, but results of algorithmic detection of temporary water coverage. We are currently undertaking a reanalysis of past floods with use of hydrodynamic modelling, and employed the Global Flood Database for verification of modelled flood extent. In the process, we have found very large discrepancies between impacts based on satellite footprints and reported impacts. For the UK, the Global Flood Database indicates an order of magnitude higher number of persons affected than detailed post-flood government surveys. In other countries, the extent of flooding is vastly underestimated, showing even two orders of magnitude lower number of persons affected. We plan to include the comparison in our follow-up study, while for now we consider that our data should be used to validate satellite-derived footprints, rather then the other way around. In this context, we didn't use spatial footprints from DFO in the database, only quantitative and textual information contained in it.

**Comment 2:** How can you differentiate between riverine and compound floods in delta areas? Are you comparing coastal and riverine floods that occurred in the same area at the same time?

**Response 2:** Yes, the events in deltas were considered compound, if there was a storm surge coinciding with a high river flow. We provide the details in section A.7:

"Compound floods were recorded as "River/Coastal" type to avoid confusion as to the meaning of "compound" flood. This type was assigned only if the description of the event indicated that both riverine and coastal flooding contributed to the impacts. If high river flows and a storm surge merely coincided and only one of the two has caused the vast majority of losses, the event was assigned to "Coastal", "River" or "Flash" type."

**Comment 3:** Is there a reason why 2020 has a limited number of reported flood impacts?

**Response 3:** 2020 includes 30 events, which is below the average for previous decades (about 40 per year), but more than e.g. 2011, 2012 or 2017. We have modified the graph by adding 2020 to the previous decade, to avoid confusion.

**Comment 4:** Figure 3 is quite dense in information. Is there a way to remove the pie charts and make them as a separate figure?

**Response 4:** we have split the figure into two for better clarity of information.

**Comment 5:** Is Figure 4 referring to the old or new version of the HANZE database? If yes, refer to HANZE v2.1 as the new version and HANZE v1.0 as the old one within the paper. It isn't easy to understand if you are referring to the new or old version

**Response 5:** we have clarified in the figures and text that we refer to the new version only in the results. In the paper, results of v1 are only shown in section 4.2.

**Comment 6:** In section 2.1.1, are you also excluding flooding due to levee failure in addition to dam failure? If yes, how can you verify that those flood impacts were not due to levee failure?

**Response 6:** levee (dike) failures are a typical component of flood events, therefore they are included. Dam breaks can similarly be part of flood events. What we explicitly exclude, are "floods caused entirely by dam failures unrelated with a severe meteorological event", as we write in section 2.1.1. This distinction means that we include dam breaks with a hydrological cause (reservoir overflow due to swollen river), but exclude purely geotechnical failures (like the notorious Stava valley tailing dam failure in 1985 that is often included in catalogues major floods, as we show in Table 3). One can also imagine dike failures unrelated to high water levels, as happened in the Netherlands due to the 2003 drought and heat wave. Similarly, dikes and dams were many times destructed on purpose to flood certain areas during World War II (carried out by both Axis and Allied forces). Such unusual occurrences without a hydrological driver are not included.

**Comment 7:** Can you explain the differences between HANZE 2.1 and 1.0 obtained in Portugal?

**Response 7:** we already explain the difference in section 4.2: "originally most footprints were based on Zêzere et al. (2014), who indicated them only by upper-level administrative districts. In HANZE v2, the footprints were revised completely using detailed geocoded impacts by Pereira (2017)." It is therefore entirely a result of incorporating better information compared to what was previously available.

**Community comment by Olga Petrucci**

We would like to thank Olga Petrucci for taking the time to analyze our paper and data. Unfortunately, we have not been able to reproduce the statistics quoted by the author, or find the event-specific problems mentioned in the comments. We have all individually checked the data, paper and the website; as we explain below, the author most likely has gotten the wrong impression of the data by analyzing only a small portion of the dataset.

**Comment 1:** Your DB, for a period of 128 years, in 42 European countries, reported 704 fatalities, while one of the papers that you quoted (Papagiannaki et al, 2022) reported 2,875 fatalities "from 12 territories in (nine of which represent entire countries) in Europe and the broader Mediterranean region" in 40 years. Obviously, a more precise comparation could be easily done, because several countries are included in both HANZE and Papagiannaki et al, 2022.

**Response 1:** The dataset covers a period of 151 years, of which in 150 separate years there is at least one event, and at least fatality. The total number of fatalities of the 2521 events is 19,322. We don't know what subset of HANZE was used by the author to arrive at 128 years and 704 fatalities. We should also note that though the other dataset (FFEM) has 2875 fatalities (as opposed to 3455 in HANZE), almost half of it is in Asian countries not covered by HANZE (Turkey and Israel). We added a comparison with FFEM in section 4.1.

**Comment 2:** Moving to the national scale, if for Italy we compare fatalities in the same time span reported in HANZE and in Papagiannaki et al, 2022, we found 102 vs 425 fatalities! There must certainly be a reason explaining this large difference…

**Response 2:** For Italy, 1980-2020, HANZE includes 625 fatalities, therefore we again think that the author only looked at some subset of HANZE.

**Comment 3:** As you said, the majority of Italian records came from a source labelled as "CNR (2023)". Actually, you mean the database of the Project AVI (Aree Vulnerate Italiane), realized by IRPI (Istituto di Ricerca per la Protezione Idrogeologica) of CNR (Consiglio Nazionale delle Ricerche). AVI project and its inventory ended around 2000, and then the events occurred after that year are not reported in it. Then also your sentence "Finally, several countries maintain national databases… (Guzzetti and Tonelli, 2004)" is not correct, because AVI DB is not updated to nowadays.

**Response 3:** We use "maintain" in the sense that they are currently accessible online, as most of them aren't actually updated (regularly at least). We did realize that event-specific data in AVI end in 2001 and some summary data in 2003; we used heavily the environmental yearbooks published by the Italian government afterwards. We modified the text so it now reads "…several countries maintain or have maintained national databases…".

**Comment 4:** Moving to the regional scale, for my region (ITF6) a very large number of papers have been published about major floods, with or without fatalities. Nevertheless, focusing on fatalities, in HANZE I only found 74 fatalities and no trace of catastrophic events killing hundreds of people in autumn 1951 and 1953, or 6 people in 1996 in Crotone or 12 in 2000 (Soverato) and 10 victims in Raganello flash flood in 2018…I know that place and related exact number of fatalities is a very local kind of knowledge that is very difficult to find looking at the European scale.

**Response 4:** Events in HANZE that have at least partially occurred in Calabria have 429 fatalities. None of the examples of supposedly „missing" floods is correct. The 1951, 1953, 1996 and 2000 events are all in HANZE, as can be quickly checked through the website:

https://naturalhazards.eu/details,16134

https://naturalhazards.eu/details,16164

https://naturalhazards.eu/details,16512

https://naturalhazards.eu/details,16527

As for the 2018 event, it was specifically excluded as mentioned directly in the paper (L508-L511 of the original manuscript): "certain flood-related cases of fatalities were excluded here: … Fatalities related to sports activities in remote areas, such as canoeing or hiking, in connection to sudden localized flash floods, as they cannot be reliably modelled both in terms of hazard and exposure, e.g. … Italy 2018 (10 fatalities, FloodList, 2023)"

**Comment 5:** Nevertheless, there is a real risk that this region is considered almost flood free, if compared to other for which you research has been more accurate. This result can remain in an "academic" environment, affecting more or less theoretical studies, but it could also be used at the European level to plan polices and investments for flood impact reductions: in this case, the use of these data can have consequences in terms of investments.

**Response 5:** As evidenced above, we don't imply that the region is "risk-free". Even a look at Fig. 4, or our online map https://naturalhazards.eu/map, gives the exact opposite impression. We mention several times the limitations of the data, and as the study doesn't say anything about future impacts of climate change in particular, it can't be used directly for any planning purposes by any reasonable decision-maker.

**Comment 6:** Finally, I have some doubts on the possibility of a correct separation between flood and landslide victims, particularly for older events, and especially if using international data sources. Nevertheless, in your paper, I did not find an explanation of how this can be exactly done. In some case, this kind of mistake is "inherited" from one DB to another. As an example, the Dartmouth Flood Archive wrongly classifies 160 victims of the 1998 Sarno mudflow in Campania (Italy) as flood fatalities. HANZE classified Sarno mudflow event as a "flash flood" but without fatalities.

**Response 6:** It is not true that we show no fatalities for the 1998 Sarno disaster. The event was indeed mostly a landslide, but the AVI catalogue attributes a very small share of impacts (1 fatality, 35 affected) in Caserta province to flood. Therefore, we included those impacts as such, while describing the full impact of the landslide in the "Notes" field for context: https://naturalhazards.eu/details,16517 . Also, we mention several times the problem of co-occurring hazards and the fact that they cannot always be separated from floods (e.g. L131-140, L374-384, L565-582 of the original manuscript). In the data, we indicate co-occurring landslides or mudslides for 227 events (in "Notes" field).

**Comment 7:** Then, in my opinion and for my experience, a DB like HANZE, virtually cannot be considered complete, and the few comparisons that I reported in this note could induce at least to reformulate one of the sentences of your paper: "However, the largest resource on occurrence of past damaging floods in Europe remains the HANZE (Historical Analysis of Natural HaZards) database (Paprotny et al., 2018a). It contains 1564 events, covering 36 countries and the period from 1870 to 2016, based on more than 300 data sources".

**Response 7:** Given the evidence above, the comment is erroneous as it is based only on a fraction of our data. Further, we didn't claim that our work is complete, but the opposite: we indicate the issue of incompleteness several times, including just before the results (L238-240 of the original manuscript), extensively in the discussion, and in the conclusions.